# The Illustrative Understanding on the Informal Sector and Its Influence in Firm Productivity in Sub-Saharan Africa (SSA): Evidence from Cameroon

**Nguepi Tsafack Elvis [1], Hua Cheng [1,\*] and Buregeya Ingabire Providence [2]**

[1] School of Economics and Management, Zhejiang Sci-Tech University, Hangzhou 310018, China
[2] College of Textile Science and Engineering, Zhejiang Sci-Tech University, Hangzhou 310018, China
\* Correspondence: chenghua@zstu.edu.cn; Tel.: +86-138-5809-2470

**Abstract:** As globalization continues, textiles and clothing firms have many opportunities around the world. Using survey data, we evaluate the determinativeness of the informal sector and its impact on firm productivity by applying the Hicks–Moorsteen index and Data Employment Analysis (DEA) methods. Specifically, this study estimates factors driving total factor productivity (TFP) and its constituents for Cameroon's companies from 2005 to 2014. As a result, the input levels of informal textile companies are the significant drivers of TFP with a mean in productivity of 47.06% for textile and 56.69% for clothing (I). Regarding formal textile companies, technical efficiency and technological progress fluctuate throughout the period of study (II). Firm size, technology adoption and primary resources (raw material) are important stages of a firm's decision to innovate (III). Therefore, employing this approach could be reliable in analyzing firm productivity in other SSA countries.

**Keywords:** informal sector; firm productivity; textiles and clothing firms; Hicks–Moorsteen index; firm size





## 1. Introduction

Developing economies are widely categorized by massive informal economy sectors, consisting of non-productive firms and providing low wage jobs [1]. The informal sector encompasses units involved in the generation of goods and services with the aim of creating jobs and wages to the people in order to earn a living. As previously described, these units mainly function at a low level of organization with a negligible or absence of division between employer and capital, which are the main features of production [2]. Recently, the informal sector, usually consisting of service suppliers, home-based workers, and unpaid workers in some enterprises, is also defined as a part of economy that is neither taxed nor supervised by any governmental institution [3–5]. From this, the informal sector is a reliable part of the economy and especially of the labor market in developing countries through enabling job creation, production, and income generation [6]. In the report of African Development Bank Group in 2015, the informal sector promotes around 55% of SSA's Gross Domestic Product (GDP). Mainly, small and medium-sized enterprises (SMEs) are frequently apprehended as the key inputs of economy growth [7], technological advancement, and regions development, to prevent from increase in jobless rate, and lead to more than 85% of turnout and jobs in most African countries [8]. The prevalence of informal companies suffers from low productivity, importer of primary resources, technology adoption, unskillful labor productivity, and high-levels products. In many developing countries from Africa, Asia, Europe, and Latin America, the informal economy sector is where more than 60% of the world's employed population are involved and where the majority of the population are, including most of the poor, principally young employees and women [9,10]. For example, firms' productivity in Africa is forced by several elements,

namely limited access to finance, unstable intermediate inputs such as poor infrastructures, energy consumption, transportation, technology adoption (by the lack of better use of machinery, servicing, and repairing skills), and business management [8]. This sector is indispensable for the fight against poverty, which is at the heart of the concerns of developing economies' policies.

The informal sector acts as the most important portion of many sectors across the African continent, mainly in construction, commerce, finance, and mining. Business activities, as well as street hawkers, are vast ordinary forms of business in Africa's informal economy sector. The sector contributes about 75% of employment [11] and nearly 70% of non-agricultural employments, about 78.6% if South Africa is excluded [12,13]. For SSA countries, the informal sector is not new. As a matter of fact, the kinds of activities executed in this sector have existed even prior to imperialism. Later, independence carried with it the distinction between informal and formal businesses, as countries around the region requested to regularize or "innovate" their economies. Within SSA, Cameroon is well correlated in view of the preponderance of the informal sector in the economy; the International Monetary Fund (IMF) emphasizes, in this section of African countries, that the preponderance of the informal sector in the GDP ranges between 20% and 65%, with tips observed in Tanzania and in Nigeria, the second largest economy in Africa [14].

Cameroon, with a population of over 25 million inhabitants, bordering with the second largest economy in Africa, Nigeria, also shares its borders with five other countries: Equatorial Guinea, Gabon, the Republic Democratic of Congo, Central African Republic, and Chad [15]. Since the financial crisis in the mid-1980s in Cameroon, the informal sector achieved an overriding space in the labor market and enhanced the main source of enrollment [16]. The determination of focusing on industrial regularization in Cameroon is mainly due to the large size of the informal sector recruiting the bulk of the employed population and the constructive correlation between the regulation of employment and enterprises [16]. The informal sector in Cameroon is the major job provider, with 90.5% in 2010 and 86.4% in 2016, according to the National Institute of Statistics (NIS), and 90% in 2020 [14]. The industrial sector comprises 34.1% of the informal sector. It absorbs 38.9% of informally employed labor. Textile and clothing firms comprise 5.7% of the producing enterprises of the hidden economy. The operations of this sector are primarily the manufacturing of textiles and the industrial capacity of clothing, which are within the five-growth carrying and job-producing industries detained by the authorities with the view to stimulate global economic and business expansion [17]. Within ten years, the country endeavors to create 45% of the regional cotton production. In the long term, the state aims to enlarge the country's cotton productivity to 600,000 tons a year by 2030 [18]. In Cameroon, the textile cotton clothing sector is, regionally, in a comparatively better position to convert cotton textile into the mass production of clothing and in circulation. These are the third and fourth stages in the flow of the textiles and clothing value chain sector. In a context of the textile clothing industry, cotton is of substantial economic importance in the northern part of the country, the poorest region. This makes Cameroon the fifth largest exporter of cotton fibers in SSA behind Mali, Benin, Burkina-Faso, and Côte d'Ivoire, and twelfth internationally [19]. Recently, cotton produces income for 2 million people (around 30% of the rural population), thus contributing, together with cereal crops grown in rotation, to food security [19]. Figure 1 shows the system of factors taken into account with the flow of information at the various stages of the value chain in Cameroon. The network of products performed by the fabrics firm is diverse: lint cotton, threading, artificial fiber, undyed, dyeing, sponge textiles, apparel, domestic washing (mantel, duvet, drapes, etc.), hose items (males, females and offspring undergarments, etc.), hats, and bags. Additionally, lower quality goods are produced by independent dressmakers and artisans [19].

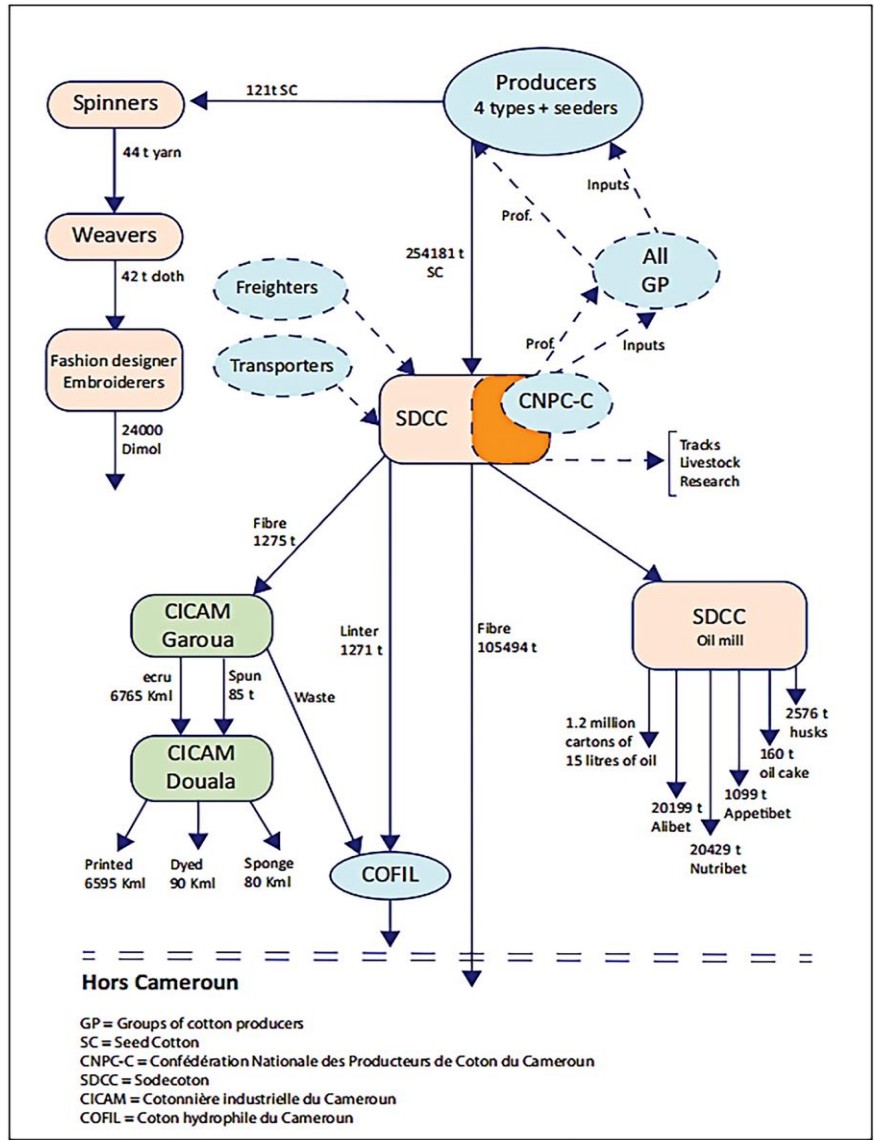

**Figure 1.** Flowchart of the cotton value chain in Cameroon. Source: Authors' compilation.

In view of this, the informal sector could be seen as a continuity between more adventuring low capacity businesses, one the one hand, and effective higher production activities, on the other hand. For intervention purposes, there are still some open challenges about the informal sector in SSA industries: First, is firm productivity in the informal sector lower than in the formal sector? Second, what are the values of returns to TFP of textiles and clothing industries? Finally, is there a relationship among firm size, TFP and its various constituents? These are among the difficulties that we endeavor to answer in this research.

The Cameroonian textile and clothing industry is an intriguing case in several areas: First, areas such as textiles are often the "majoring sectors" because, as a country begins to develop, these sectors entail moderately lower capital investment in view of primary resources and human capital [20]. Second, the Cameroonian cotton textile sectors ranks third amongst productivity firms with a share of 9.5% in the GDP, behind Mali with an annual production of 23,085 tons and Chad with an annual production of 48,821 tons in 2019. Figure 2 shows the top five SSA countries producing (2019) the cotton supply [21]. This shows that SSA is a key player in the supply of sustainable cotton. The World Cotton Day 2021 reported that, for the 2019 agricultural campaign, out of an annual productivity of 138,585 tons of cotton manufactured in Cameroon by the Cotton Development Company (SODECOTON), about 5% of this cotton was converted nationally. According to the special-

ists of the sector and the government institution, textile and clothing firms demonstrated a fast decrease in their production, with the exception of the section of cotton wool, which has the largest in the capacity of the sector. Therefore, Cameroonian institutions, supported by the World Bank, are involved in a specific project founded on the national innovation and assessment of cotton fiber output by 2035, in order to revitalize the sector and to secure a national and global marketplace [22].

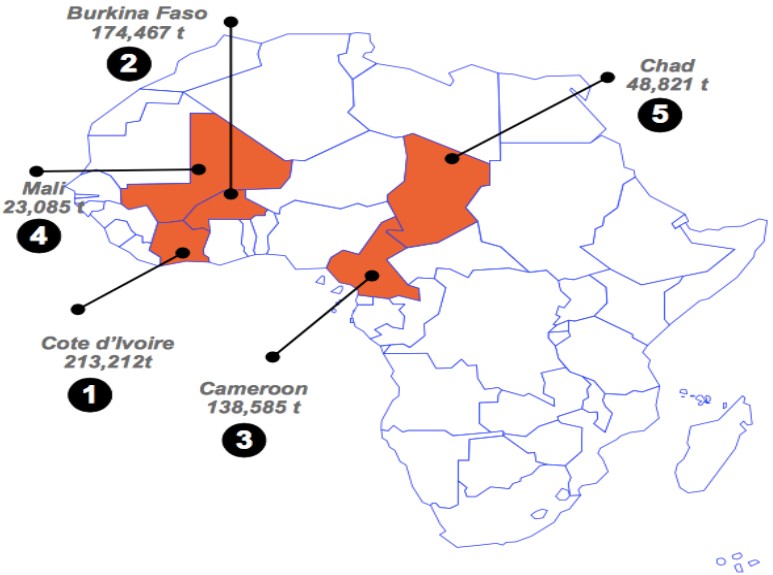

**Figure 2.** Top five SSA countries' manufacture. Source: Textile Exchange [21].

Lastly, according to World Bank reports, textiles and apparel have experienced tremendous growth from 2017 to 2018 [23]. With the COVID-19 pandemic to blame, world exports of apparel decreased from USD 492 billion in 2019 to USD 448 billion later in 2020 [23,24]. Guided by the increase in personal protective equipment (PPE) productivity, worldwide textile exports increased by 16.1% in 2020, achieving USD 353 billion. The universal commodities businesses in 2020 also experienced an unusual 8% decrease during the past year [24]. Figures 3 and 4 report the World Textile and Apparel Trade data for 2020. This poses a duality, with rising earnings in China, the world's largest textile producer and exporter [24], along with providing tremendous scope for countries such as SSA (i.e., Cameroon, Mali, and Burkina-Faso) to enhance its quota in worldwide textile exports [25].

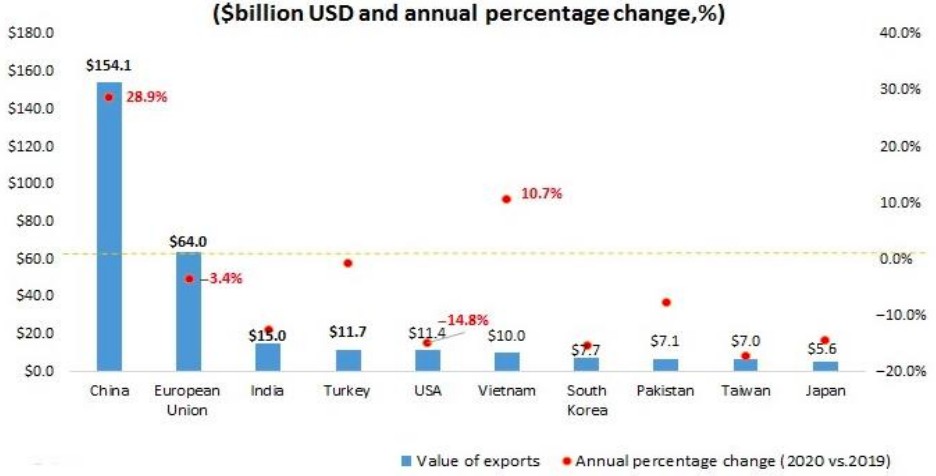

**Figure 3.** Top ten exports of textiles in 2020 (USD billion and annual percentage change, %). Data source: [24].

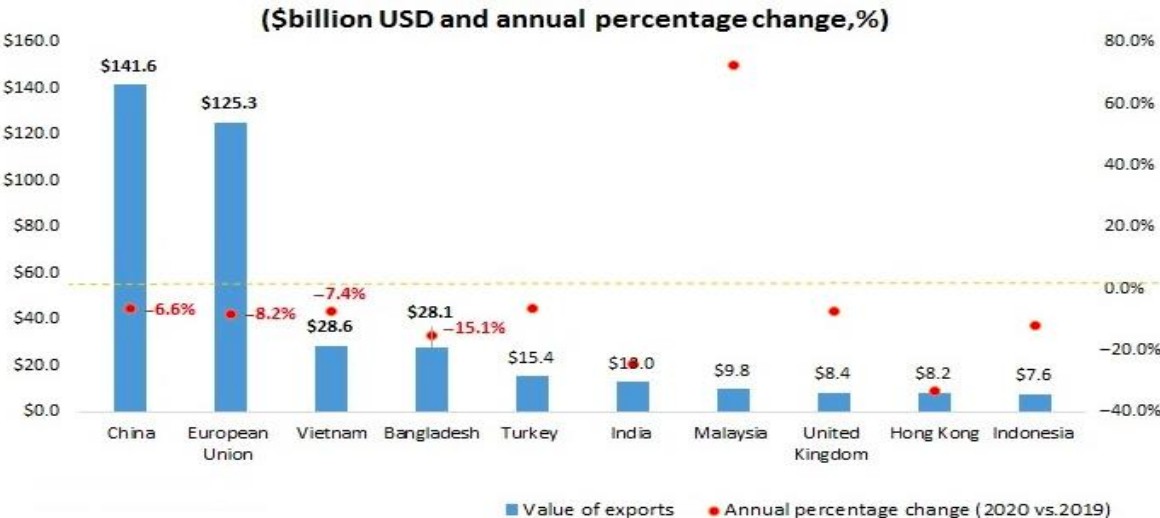

**Figure 4.** Top ten exports of clothing in 2020 (USD billion and annual percentage change, %). Data source: [24].

From what we know, this research is promising by incorporating the Hicks–Moorsteen index with the DEA-based method, which may lead to further research to analyze other empirical surveys on informal sector productivity analysis on textile and clothing industries in SSA countries, especially Cameroon; however, most of the existing informal sector-related literature strongly highlights the Malmquist index. The Hicks–Moorsteen index provides sufficient details on productiveness and meets the conclusiveness property under vulnerable conditions on applied science. Consequently, this applied framework leads to viable final results under variable returns to scale compared to the Malmquist index. The relationship between the informal and formal sectors is displayed in Section 3. Relying on the merits of this method, we used data from the survey of the Employment and Informal Sector (EESI) carried out between 2005 and 2010, and General Enterprise Survey (RGE) performed in 2009 and 2014 of the textile and clothing industry sector in Cameroon. In this study, we report the first analysis of how factors such as primary resources, technology adoption, and firm size have an important role in deciding TFP in Cameroon. For example, considering the firm size data, we found that labor and being capital intensive are the most significant factors of production affecting TFP in the manufacturing industries of SSA countries. The results may be useful for Cameroon but also for other SSA countries. The rest of this paper is structured as follows. Section 2 reviews the literature. Section 3 provides an overview of the informal sector and its determinativeness. Section 4 describes the methodology. Section 5 presents the DEA and Hicks–Moorsteen index models. Section 6 discusses the results of the TFP and its constituents. Finally, Section 7 condenses the main findings of the research.

## 2. Literature Review and Research Hypothesis

### 2.1. Theoretical Framework

The review of the literature on the evolution of firm productivity of textile and clothing industries shows that employing the Hicks–Moorsteen index has been disregarded in SSA [26–29]. In line with previous investigations from developed countries, TFP of the textile and clothing industries is mostly reported regarding the Malmquist index. As introduced by [30], the technology-based and time-discrete Malmquist productivity index builds a production frontier illustrating technology and employs distance functions estimated at different input–output compounds for production comparison. Ikasari et al. [31] investigated the productivity of the textile industry in Indonesia between 2010 and 2011 using the Malmquist index. They discovered a growth of production of 1.6% explained by technological change. Kapelko and Lansink [32] examined the productivity of the textile

and clothing industries around the world between 1995 and 2004 using the Malmquist index. They noticed a relatively low growth in production in the sector because of the increase in technical progress.

All these researches investigate firm productivity of textile and clothing companies employing the method of the Malmquist index to estimate the growth in productivity and its several constituents. Nevertheless, in defiance of the obvious popularity of this approach as a measure of change in productivity, the effect of the estimates of Malmquist's index furthermore assumes constant returns to scale questionable [33].

In view of this observation, the Malmquist index conducted by Fare et al. [34] is a prejudiced measure of the TFP in the absence of the hypothesis of constant returns to scale. Therefore, the decomposition of this approach suggested by Fare et al. [34] leads to not very dependable estimates of technical efficiency change and technology shifts, as described by Ray and Desli [35]. Bjurek [36] explained that the Hicks–Moorsteen productivity index provides detailed information on TFP interpretation and Grifell and Lovell [37] and Färe et al. [38] demonstrated this mathematically. In addition, with the formerly established doubts of the Malmquist index as a TFP index and the potential infeasibilities it may be affected by, one favorable way to understand this is through the use of the Hicks–Moorsteen index.

### 2.2. Research Hypotheses

With the influence of the informal sector on the measurement on TFP, there are diverse responses [39]. The formal sector imposes national firms to pay taxes for environment protection, as a result of which firms acquire pollution protection equipment or reduce their environmental pollution by funding high technology [40]. To save costs, firms may choose to enlarge their capacity scale.

In terms of the time tendency of the TFP industry, the economic conditions and political situation play a significant role. For instance, the disruption of the economic crisis in 2008 and 2020 led to a long-term repercussion on the nationwide economy and thus a lower technology of firms, which in turn decrease the TFP. Government institutions in charge of the informal sector noticed the potential consequences on TFP. In addition, TFP and its constituents' contributions to the firms, and the situation of the scale-mixed economy are coordinated matters since the constituents of TFP differ for different sectors. Specifically, sectors with a harmonious development and relatively mature and complete manufacturing distribution have the advantage in employing assets and developing sustainable outputs. In accordance with the above discussion, we introduce the following hypotheses and test them in Section 6.

**Hypothesis 1 (H1).** *TFP and its constituents: technological progress (TECH) and technical efficiency (TE) are concurrently affected by the determinativeness of the informal sector and government policy. (TECH) and (TE) have negative impacts on TFP.*

**Hypothesis 2 (H2).** *The efficiency-mix and scale have no impacts on the change in TFP, which negatively affect firm productivity of textiles and clothing industries in Cameroon during the period 2005–2014.*

### 3. The Informal Sector and Its Determinativeness: An Overview

#### 3.1. Overview of the Informal Sector

In African countries, the informal economy sector largely contributes to the Gross National Income (GNI). First, the informal sector enables developing countries to prevent the increase in the joblessness rate and to propagate the revenue of the predominance of locals, including the provision of important assistance to the lower segments of the community, regarded as an additional benefaction to the formal sector [41]. Research conducted in African nations and African Development Bank (AfDB) in 2018 showed that the significant average of the informal sector to the GNI for Africa, South of the

Sahara, is 40.2%, two-times lower than that of Central Africa at 91.0%, Eastern Africa at 91.6%, and Western Africa at 92.4%. As demonstrated, sub-region workers act for 84.3% of total employment in comparison to 37.2% for SSA [42]. Second, the cross-analysis of the informal sector in six countries reveals that the sector is a fundamental stage of progress to the marketplace and business for the bulk of youthful employees and adults in Africa. The sector absorbs up to 35.3% of South Africa workforce, and nearly 95% of Benin, and about 90.2% of Cameroon (AfDB: 2018). This is further illustrated in Table 1.

**Table 1.** Size of the informal sector according to the increase in the joblessness rate.

| Country | Measure of Informal Sector Employment | Measure of Employment in Companies | Measure of Employment in Administration |
|---|---|---|---|
| South Africa | 35.3% (all sector) | 69% | - |
| Côte d'Ivoire | 92.8% (all sector) | 6% | 5.6% |
| Benin | 95% (all sector) | 5% (administration included) | - |
| Cameroon | 90.2% (all sector) | 4.7% | 4.9% |
| Ethiopia | 90.8 % (all Sector) | 6.2% | 2.6% |
| Senegal | 90% (all sector) | 16.8% | 5.7% |

Source: [41], Harmonized series ILO and Bonnet, F. ILOSTAT database, 2018.

In Table 1, the size of informal employment compared to overall employment shows that SSA is expected to have more people entering the workforce in African countries such as Ethiopia and the four Francophone SSA countries than other continents [16,41,43]. An increase involves a higher need for employment than the public employment is probably capable to provide. In SSA countries, 89.2% of employers approximate 20% age points higher than in developing economies and emerging markets are engaged in informal employment, with significant nationwide changes [16].

The informal sector shows that the shadow economy acts for business activities by private operators and business units having informal legal agreements [44]. The informal sector acts by excluding unlawful activity operations as reported by the seventeenth International Conference of Labor Statistics (ICLS) [45]. These businesses activities intervene outside the accepted norms of society in that they are not officially registered with the government and are hence non-taxable. In addition, [45] postulates that employers in the informal sector usually operate at the economical level of organization and on a small scale. Work relations, when existent, are based on employers and employees' relationships contrary to contractual arrangements with legal safety. Table 2 describes the main points of distinction between the formal and informal sectors, such as the differences on the basis of entry, the skills and qualification, the legal status scale of operations, pay structure, organization structure, regulation, authority and hierarchy and competition [46].

**Table 2.** Differences in the relationship between the informal and formal sectors.

| Informal Sector | Formal Sector |
|---|---|
| Entry demands are minuscule/easy | Difficult entry demands |
| Rely on local/indigenous | Frequently relies on foreign resources |
| Ownership family companies | Ownership is through a corporation |
| Business operation is a small scale | Company operation is a large scale |
| Operation is labor intensive, using locally adapted technologies | Operation is capital intensive depending on imported technologies |
| Unskilled employees acquired outside the formal school system | Formally acquired skills, often the use of expatriate services |
| Generally competitive and unregulated markets | Markets are protected by the use of high tariffs |

Source: [46,47].

### 3.2. Informal Sector and Its Priority in Cameroon's Economic System

The informal sector accounts for insignificant (very small), unrecorded private business operating on a small scale outside the agricultural sector and whose firms enable them to produce at least part of what is on the market [48]. Especially, the informal sector incorporates businesses that do not contribute to government accounts. The size of the informal sector in Cameroon can be evaluated both at the macroeconomic and microeconomic levels. At the macroeconomic level, the sector represents more than 57% of the country's GDP. This benefaction is below that of SSA at 63.6%. On the other hand, it is above the benefactions of Asia and Latin America, correspondingly estimated at 29.2% and 30.2%. Based on [48,49], the contribution of the informal sector to the GDP, excepting agriculture, follows the same trend. Thus, we have 36% in Cameroon as against 14.2% in Asia and 24% in Latin America. At the microeconomic level, Cameroon has more than 2,500,000 Informal Production Units (IPUs) throughout the country [11] (National Institute of Statistics–NIS, 2011). In terms of distribution, rural areas account for 49.5% of these units, while metropolitan areas, the cities of Douala and Yaoundé, constitute 33.3% of IPUs. In terms of their management, 54.4% of IPUs are managed by young workers and women, while men manage only 45.6%.

### 3.3. The Results of COVID-19 on the Informal Sector in Cameroon

The COVID-19 pandemic presented problems and opportunities of the informal economy in Cameroon. The challenges faced by the informal economy sector in Cameroon as a consequence to the coronavirus pandemic are diverse. Indeed, the public administration took several measures to strengthen business and domestic activities to support them in the management of the outbreak [50,51]. The NIS records concerning the conditions of the outbreak in Cameroon specifies that about 80% of entrepreneurs in the formal sector and 82% of informal unit heads believe an average and significant slowdown in the sector [52]. Various firms have experienced a layoff on their business activities, including about 13% of firms in the informal sector and nearly 15.7% of firms in the formal sector, with 5.4% for large firms and 10.3% for SMEs. Nearly 82% of firms indicated a drop in their manufacturing of around 50% [53], and more than 95% reported a downturn in their income and gross revenue, and more than 58% decreased the quota of their workers [52]. Several response policies were applied by firms, especially the depletion in working time, the suspension of prepared investments, the nullification of orders from dealers (50.1% of firms), the temporary shutdown of certain workers, the moderation of earnings as well as the implementation of working from home [52]. Subsequently, the ratio of informal jobs in the economy sector rose with the inflation of subsistence businesses.

### 3.4. The Determinativeness of the Informal Sector in Cameroon

The surplus of Cameroon's informal sector is fivefold. First, the daily per hour work in the informal economy stands at CFAF 463 (nearly EUR 0.70). It extends to a maximum level of CFAF 1037 (nearly EUR 1.60) for IPUs with three people and falls beyond that [54]. The challenges of the degree of schooling and the degree of information guidance clarify to some extent the rate of the informal sector. Second, although low production is the main constraint for the regularization of enterprises, it remains that the extreme progress of this sector also arises from insufficiencies related to economic governance. Indeed, the economic adjustment system still affects the shame of its emigrant sources. Third, the limited access to finance conjointly performs a leading role in the economy sector since most firms lack business assets and venture capital, which makes it difficult for them to obtain funding. The lack of banks in rural areas creates access to monetary services impracticable. Accessible finance could create a massive impetus for the informal economy sector to join the formal economy. Fourth, the limited market information means that it is tough for informal economy cross border business activities to comprehend regional and sectoral commercial prospects. Across the regions, the information provided on cross-border businesses is, at best, essential. Since most firms conduct outside repetitive trade communities, market information on prices, supply, and consumers is usually not available to them. They

generally depend on the informal economy sector, at times deceptive information networks, guidance on policies, regulations, agreements, and protocols for the benefit of the sector. Fifth, management skills are also at the essence of addressing the sector issue. The informal economy sector is essentially less educated and often lacks basic trade management skills. Trade skills are mainly acquired via traditional means and numeracy, including literacy levels, which are notably low in some localities.

The informal sector, viewed as a refuge sector for many people finding it difficult to combine the formal constitution, protects from policy-making persistence. Additionally, the indulgence, the absence of openness, and scarcities of the public sector service encourage the maintenance of an important segment of productiveness units in the informal economy [55]. This is further illustrated in Figure 5.

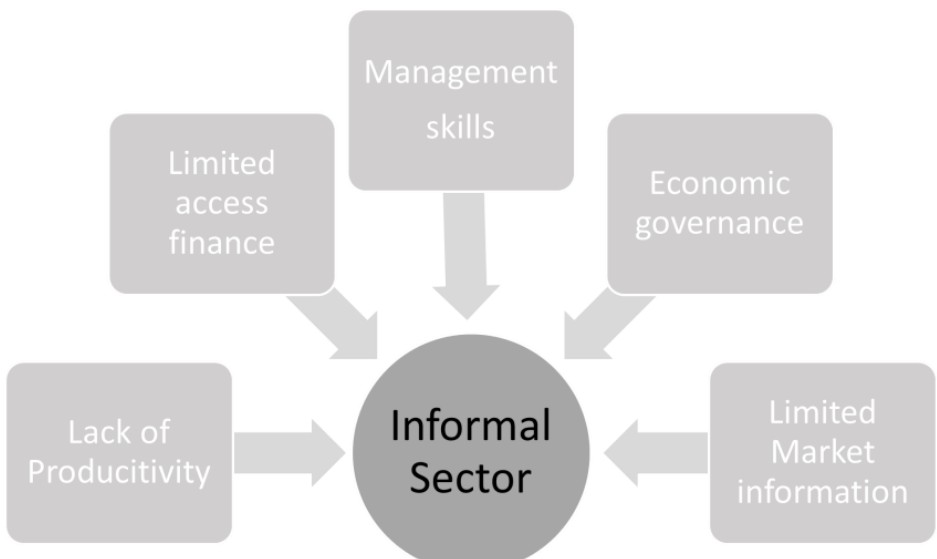

**Figure 5.** The key determinants of the informal sector in Cameroon. Source: Authors' compilation [55].

## 4. Methodology: Specification of the Technology

In this section, the study evaluates the benefits of the informal sector to firm productivity of SSA countries through a systematic review of papers from the literature. Additionally, we examine the specification of information technology (innovation) to which the Hicks–Moorsteen index is computed. The DEA-based and Hicks–Moorsteen index methods were implemented to study the causes of operational efficiency and dynamic change in the databases of the study period. Additionally, the variable estimation of the constituent efficiency estimates of the Hicks–Moorsteen index was tested.

### 4.1. Data and Variable Description

4.1.1. Data Description

The two databases employed in this survey derive from the NIS of Cameroon. Firstly, the database on the RGE was performed in 2009 and 2014. We used the dataset on the firm size of the textile and clothing industries. From these yearly statistics, the industry sector has 11,685 enterprises and is distinguished by a frequency of very small enterprises (VSE) in numbers of 9917 enterprises, with a percent of about 85%. Major enterprises represent over 2%. The textile and clothing industries control and have 54.7% of the enterprises [56,57]. Secondly, the EESI as regards the non-agricultural informal sector, fulfilled between 2005 and 2010 by NIS, was used to evaluate firm productivity in the textile and clothing enterprises. Regarding the informal sector in Cameroon, around 4592 informal firms were studied [17,58]. These two databases were studied to evaluate the effect of the role of firm productivity of textile and clothing enterprises.

To evaluate the TFP of IPUs of the sub-industries, the study on EESI allowed us to recognize a selection of 428 informal textile and clothing industries units of production, 102 being informal textiles units of production and 326 being informal clothing units of production [17,58]. The formal sector classifies a sample of 278 formal enterprises of textile industries into three sub-groups [56,57]. The first group is that of VSE and consists of 153 in the textiles industries. The second group accounts for small enterprises (SE) and retains 83 in the textile industry. The third group is composed of medium enterprises (ME) and has 42 in the textile industry [56,57].

4.1.2. Variable Description

The ordering of the relevant inputs and outputs is a major decision in the calculation of firm productivity [59,60]. Surveys on the productiveness of the textiles and clothing industries generally use capital and labor intensives as inputs, while the quantities manufactured constitute the outputs. In addition to the direct input of the sub-industries (textiles and clothing), TECH is an important factor for improving textiles and clothing industries efficiency. As concerns sub-industries, studies have appraised the mass production of Indian clothing firms by employing the pieces of clothing manufactured as the number of outputs produced and the embroidery machines and tailors or clothiers as the number of inputs [61,62]. Hussmanns [63] estimated the mass production of firms in the textile and clothing company in India. The annual turnover was employed as the output, whereas the number of workers, the number of embroidery machines, and the consumption of energy were applied as inputs. The authors of [64] conducted their investigation in the manufacturing industry of textiles in India. They had three inputs and one output. The output assigned was the total value of the products of the firm during the year. The input wase labor measured by the total number of working days per worker. The amount of working time (hours) is a good selection as a workforce input [65]. Thus, we considered that the contrary number of workers is due to the limited information on income and working time, which is comparable with [65,66]. Capital represents the value of the resources fixed from the beginning of the year and intermediate inputs (fuel, energy, and water). Ikasari and et al. [23] evaluated the productivity of the textile industry in Indonesia. They employ the cost of primary resources, the cost of labor, energy and gasoil as inputs. The textile products obtained represented the value of the output. The selections of the parameters in our survey was conducted by the former surveys above and the availability of the databases. As a result, the paper chose five inputs and two outputs. Table 3 condenses the variables employed.

**Table 3.** Variables of the study.

| Variables | | |
|---|---|---|
| | Inputs (annual) | Outputs (annual) |
| Textile firms | -Cost of the intermediate inputs (water, gasoil, transport, energy in megawatt, and cost of raw material). | -Annual production (CFAF) |
| | -Number of workers | -Annual turnover (CFAF) |
| Clothing firms | -Number of sewing machines | -Annual production (CFAF) |
| | -Number of working hours by tailor<br>-Cost of the intermediate inputs | -Annual turnover (CFAF) |

### 4.1.3. Statistical Description

Tables 4 and 5 summarize the statistics of the general information on the observations and distribution of the informal and formal sectors' firms in terms of outputs, inputs, periods, and size.

**Table 4.** Informal textile and clothing sector. Source: Employment and Informal Sector Surveys [17,58].

| | 2004 Mean St Deviation | 2005 Mean St Deviation | 2009 Mean St Deviation | 2010 Mean St Deviation |
|---|---|---|---|---|
| **Textile Inputs (annual)** | | | | |
| -Cost of the intermediate inputs (water, gasoil, energy cost, and raw material costs) | 252.7493 441.1844 | 288.7695 211.7742 | 1826.5807 4120.1120 | 1978.8709 5291.2982 |
| -Number of workers | 1.1569 0.4635 | 1.198 0.3931 | 1.441 0.2893 | 1.391 0.5698 |
| **Outputs (annual)** | | | | |
| -Annual production (in CFAF) | 513.791 818.2021 | 1252.1421 1073.2140 | 3225.1750 963.234 | 4134.2485 774.7500 |
| -Annual turnover (in CFAF) | 522.1895 827.9051 | 980.2873 1470.7231 | 3965.2418 361.7891 | 4829.3214 1236.6736 |
| **Clothing Inputs (annual)** | | | | |
| -Number of annual working time (per worker) | 1721.2 169.8 | 1616.9 202.6 | 2111.2 223.6 | 1851.4 135.3 |
| -Cost of the intermediate inputs | 418.5954 975.3989 | 724.2012 520.3230 | 1273.4120 210.3210 | 1124.7593 2006.7050 |
| -Number of sewing machines | 1.59 1.203 | 1.66 1.175 | 1.96 1.3442 | 2.09 1.470 |
| **Outputs (annual)** | | | | |
| -Annual production (in CFAF) | 988.2303 1720.6538 | 1096.3251 18,341.2544 | 1789.2341 2075.2314 | 1997.4154 3020.1855 |
| -Annual turnover (in CFAF) | 1010.1126 1869.6293 | 1855.7892 2223.1420 | 2161.4769 4454.3317 | 2046.9400 3437.1873 |

Regarding Table 4 (informal sector), on the annual turnover in 2010, the standard deviation of 1236.6736 of textile outputs is larger than the clothing company's output annual turnover of 3437.1873. However, the annual production of outputs in clothing companies of 3020.1855 are larger than in textile companies' annual production, at 774.7500. Another important observation is that the firm size of the clothing companies is large compared to that of the textile companies. This shows a greater dispersion in the distribution of labor intensity and the number of sewing machines, with a low level of technology for clothing companies.

Table 5 reports the distribution of formal textile companies in three sub-groups, VSE, SE and ME. In terms of size, VSE is larger than other two sub-groups. On average, the standard deviation on the annual production and turnover of ME in 2014 is the largest value compared to the other sub-groups. In view of this observation, capital intensity related to the acquisition of machinery and high-level technology. This finding is in line with the idea that, in developing countries, the acquisition of machinery, technology adoption, and product innovation are the most dominant innovation strategies in the formal sector.

**Table 5.** Formal textile and clothing sector. Source: General Enterprise Survey [56,57].

| Very Small Enterprises (VSE) | | | | | | |
|---|---|---|---|---|---|---|
| | **2008–2009** **Mean** **St Deviation** | **2009–2010** **Mean** **St Deviation** | **2010–2011** **Mean** **St Deviation** | **2011–2012** **Mean** **St Deviation** | **2012–2013** **Mean** **St Deviation** | **2013–2014** **Mean** **St Deviation** |
| **Inputs (annual)** | | | | | | |
| -Cost of the intermediate inputs (water, gasoil, transport, energy cost, and raw material) | 924.2 1666.33 | 905.20 1455.256 | 1075.1 2011.275 | 1124.21 2140.21 | 2088.141 1175.5 | 1175.5 2214.247 |
| -Number of Workers | 2.028 1.12 | 2.085 1.33 | 2.22 1.035 | 3.088 1.111 | 2140.21 830.2 | 3.055 1.481 |
| **Outputs (annual)** | | | | | | |
| -Production annual (in CFAF) | 2330.2 2926.3 | 1987.3 2142.45 | 1986.28 2075.325 | 2895.2 1731.24 | 1896.5 2210 | 2140.3 1974.211 |
| -Annual turnover (in CFAF) | 974.3 1732.214 | 1015.3 724.321 | 41.214 957.279 | 1054.6 9023.781 | 654.4 079.738 | 830.2 338.154 |
| Small Enterprises (SE) | | | | | | |
| | **2008–2009** **Mean** **St Deviation** | **2009–2010** **Mean** **St Deviation** | **2010–2011** **Mean** **St Deviation** | **2011–2012** **Mean** **St Deviation** | **2012–2013** **Mean** **St Deviation** | **2013–2014** **Mean** **St Deviation** |
| **Inputs (annual)** | | | | | | |
| -Cost of the intermediate inputs (water, gasoil, transport, energy cost, and raw material) | 3322.7 9129.991 | 3623.9 5511.121 | 3434.23 4723.311 | 3517 6623.021 | 3014.2 8011.247 | 3242.2 5012.32 |
| -Number of workers | 58.83 1.105 | 10.33 1.123 | 19.17 1.19 | 8.013 1.13 | 319.25 1.027 | 11.14 1.065 |
| **Outputs (annual)** | | | | | | |
| -Production annual (in CFAF) | 7586.3 17,285.23 | 6175.23 19,832.9 | 6021.185 19,471.21 | 6583 20,584.2 | 6222.2 17,624.81 | 6175 17,212.4 |
| -Annual turnover (in CFAF) | 6811.2 713.321 | 6053.32 882.4211 | 5824.654 1012.441 | 6346.7 1222.32 | 5830.3 1214.214 | 5113.2 902.321 |
| Medium Enterprises (ME) | | | | | | |
| | **2008–2009** **Mean** **St Deviation** | **2009–2010** **Mean** **St Deviation** | **2010–2011** **Mean** **St Deviation** | **2011–2012** **Mean** **St Deviation** | **2012–2013** **Mean** **St Deviation** | **2013–2014** **Mean** **St Deviation** |
| **Inputs (annual)** | | | | | | |
| -Cost of the intermediate inputs (water, gasoil, transport, energy cost, and raw material) | 13,087.2 22,101.2 | 17,435.2 17,446.71 | 1589.2 20,161.33 | 16,327.2 30,112.62 | 19,568.3 45,616.91 | 18,324.2 40,222.5 |
| -Number of workers | 20.22 6.024 | 18.11 5.084 | 19.12 70.23 | 22.14 60.041 | 23.15 8.015 | 24.81 7.032 |
| **Outputs (annual)** | | | | | | |
| -Production annual (in CFAF) | 71,532.4 12,598.35 | 90,229.2 21,579.12 | 61,558.7 21,579.12 | 61,848.6 28,640.6 | 51,935.3 21,343.24 | 81,532.4 18,414.2 |
| -Annual turnover (in CFAF) | 69,054.1 62,458.32 | 81,083.2 13,874.33 | 58,294.3 17,217.86 | 59,126.44 551,461.6 | 48,323.3 43,145.23 | 73,548.2 40,175.7 |

## 5. Model Specification

The indices of productivity in accordance with the manufacturing technological advancement and without details on prices are the Hicks–Moorsteen index and Malmquist index [34]. The Malmquist index is the most commonly used productivity index applied by imposing constant returns to scale. However, it was demonstrated that the Malmquist index is not at all times an index of TFP. Although the qualities are seen under the presumption of constant returns to scale, major defects or difficulties occur in the presence of inconstant returns to scale, which largely shows the genuine technology [33]. There is also a probability that the conclusions guide to major defects of infeasibility. Hence, this benefit intends to empirically examine the frequency of the insolvability challenge and to describe how the Hicks–Moorsteen index avoids it. The research of Bjurek et al. [36] is useful in this regard. To solve it, Bjurek et al. [36] suggested the Hicks–Moorsteen productivity index.

### 5.1. The TFP Measurement in the Manufacturing Industry

We analyzed the efficiency of textile and clothing industries in Cameroon's informal and formal sectors, principally drawing on the following concept. First, an index system of inputs and outputs was constructed as a quantitative objective standard for estimating the operating efficiency of the textile and clothing industries. Second, by virtue of the DEA and Hicks–Moorsteen index methods, the evaluation model of textile and clothing industries efficiency was formulated. Finally, the databases of the sub-industries in Cameroon from 2005 to 2014 were collected by an empirical survey to analyze the TFP and its subdivision efficiency, as displayed in Figure 6.

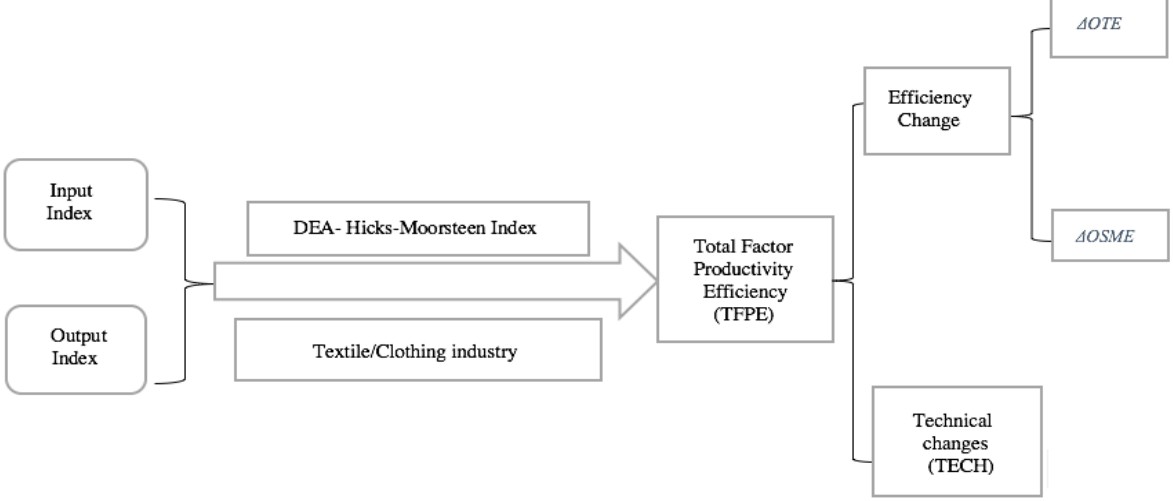

**Figure 6.** The concept of measuring TFP of Cameroon's textile/clothing industry.

The total factor productivity efficiency (TFPE) can be additionally subdivided into the technical changes index and efficiency changes index. Additionally, by introducing inconstant returns into the scale distance function, the efficiency changes the index in Equation (5), as detailed in Section 5.2. Additionally, this can be decomposed into the change in output-oriented technical efficiency (OTE) and the change in output-oriented efficiency-mix and scale index (OSME), as illustrated in Figure 6.

The DEA-based and Hicks–Moorsteen index methods were applied to investigate the causes of operational efficiency and dynamic changes in the databases of the study period.

### 5.2. General Specification of the Model

Considering a sub-industry with various inputs and outputs, [67] employed the ordinary interpretation of TFP as formulated by [68]:

$$TFP_{nt} = \frac{Y_{nt}}{X_{nt}} \tag{1}$$

where $TFP_{nt}$ denotes the total productivity of the factors of the industry $n$ during the period t. $Y_{nt} = Y(y_{nt})$ and $X_{nt} = X(x_{nt})$, where $Y_{nt}$ and $X_{nt}$ are correspondingly outputs and inputs of the productivity facility considered. Based on this conceptualization, we can define the changes in total productivity of the factors as the proportion of the index of quantity of output and the index of the quantity of the input or as the proportion of the growth of the output and the input. O'Donnell [69] referred to such index numbers as multiple-source complete. The Hicks–Moorsteen index can be evaluated without the need of price data,

which is in accordance with the works of Hicks [70] and Moorsteen [71]. The index of total productivity of the factors of Hicks–Moorsteen was edited as follows:

$$HM^{t,t+1} = \left[ \frac{D_o^{t+1}\left(x^{t+1}, y^{t+1}\right) D_0^t\left(x^t, y^{t+1}\right)}{D_0^{t+1}\left(x^{t+1}, y^t\right) D_0^t\left(x^t, y^{t+1}\right)} \frac{D_1^{t+1}\left(x^t, y^{t+1}\right) D_1^t\left(x^t, y^t\right)}{D_1^{t+1}\left(x^{t+1}, y^{t+1}\right) D_1^t\left(x^{t+1}, y^t\right)} \right]^{1/2} \quad (2)$$

where $D_0^t(x, y)$ and $D_1^t(x, y)$ correspondingly refer to the distances functions of outputs and inputs, described as $D_o^T(x, y) = \min\{\delta > 0; (x, y/\delta) \ \epsilon \ p^T\}$ and $D_I^T(x, y) = \max \{\rho > 0 : \left( \frac{x}{\rho} \right) \ \epsilon \ p^T\}$. $p^T$ illustrates the set of productivity possibilities for T periods. $\rho$ represents the maximum contraction of an input vector scalar, while $D_0^t(x, y)$ still remains on the boundary of the technology. Additionally, $\delta$ represents the minimum expansion of the output vector scalar, while $D_1^t(x, y)$ still remains on the boundary of the technology. These distance functions can be measured by employing DEA models constructed by [69]. DEA does not require any exclusive presumptions concerning the functional form and efficiency distribution. Nevertheless, DEA is a non-parametric approach, which can prevent the hindrance that stochastic frontier studies and other approaches depend heavily on assumption [72]. As a matter of fact, O'Donnell [73] expanded DEA programs necessary to measure and analyze the Hicks–Moorsteen productivity index. However, in this research, the employment of the DEA-based and Hicks–Moorsteen TFP index methods was overweighed by the feasible results and effective perception that this innovative system can contribute.

*5.3. Specification of the Hicks–Moorsteen Index*

O'Donnell [67] revealed that all indices of TFP that are multiplicatively complete, and can be decomposed into a measure of technical change and various measures of change in efficiency. We can assume the output-oriented decomposition of a multiplicatively complete TFPE index for firm n in period t and can be edited as:

$$TFPE_t = \frac{TFP_{nt}}{TFP_t^*} = 0TE_{nt} \times 0ME_{nt} \times R0SE_{nt} \quad (3)$$

where $TFPE_t$ is the maximum TFP possible employing any technical viability inputs and outputs; $0TE_{nt}$ (output-oriented technical efficiency) represents the difference between ascertained TFP and the maximum TFP that is possible while holding the input–output mix and input level settled; the output-oriented mix efficiency ($0ME_{nt}$) estimates the change in production when restrictions on the input and output mix of the industry are undisturbed; and $R0SE_{nt}$ (the residual output-oriented scale efficiency) expresses the difference between TFP at a technically and mix efficient point and TFP at the point of optimum performance.

The output-oriented decomposition of a multiplicatively complete index of TFP can be edited as:

$$TFP_{nt} = TFP_t^* \times (0TE_{nt} \times 0ME_{nt} \times R0SE_{nt}) \quad (4)$$

A correlative illustration to Equation (4) can be developed for any other industry, such as m in period s. Furthermore, the index number that equates the TFP of industry n in period t with the TFP of industry m in period s can be edited as:

$$\boldsymbol{TFP_{ms,nt}} = \frac{TFP_{nt}}{TFP_{ms}} = \underset{\text{Technical changes}}{\left[ \frac{TFP_t^*}{TFP_s^*} \right]} \times \underset{\text{Efficiency changes}}{\left[ \frac{OTE_{nt}}{OTE_{ms}} \times \frac{OME_{nt}}{OME_{ms}} \times \frac{ROSE_{nt}}{ROSE_{ms}} \right]} \quad (5)$$

The expression contained in the first parentheses on the left-hand side of Equation (5) illustrates technical changes, calculating the difference between the maximum TFP possible using any technology viable at times t and s. Consequently, the sector encounters technical enhancement or diminishment, controlled by whether $\frac{TFP_t^*}{TFP_s^*}$ is greater or less than 1. The other relations on the extreme right-hand side of Equation (5) are several

constituents of technical efficiency changes and are discussed as estimates of technical efficiency change, mix-efficiency change, and residual scale-efficiency change. We employed the DPIN software designed by O'Donnell [73] to evaluate the different estimates of efficiency and TFP index constituents.

## 6. Results and Discussion

### 6.1. TFP Results and Its Components in 2005–2014

The measurement results of TFP of the textile and clothing industries of the informal and formal sectors in 2005–2010 and 2009–2014 are shown in Tables 4 and 5 above, and Figures 7 and 8 below represent the formal and informal sectors, respectively.

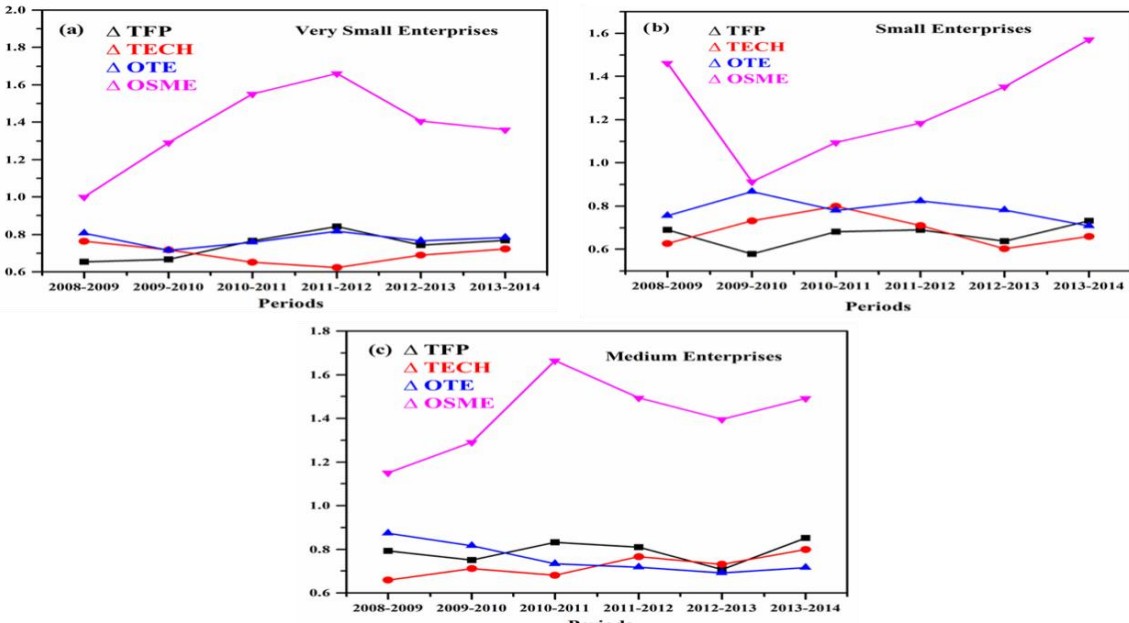

**Figure 7.** The average Hicks–Moorsteen productivity index, and its constituents at annual mean in 2009 and 2014. (**a**) Very Small Enterprises; (**b**) Small Enterprises; (**c**) Medium Enterprises. Notes: ΔOTE: the change in technical efficiency; ΔTFP: the change in total factor productivity; ΔTECH: the change in technological progress; ΔOSME: the change in efficiency–mix and scale (an efficiency value lower than 1 indicates that there is inefficiency in the year of the sector; an efficiency value greater than 1 indicates that there is more efficiency in the year of the sector).

For the formal sector, as shown in Figure 7, ΔTFP declined over the study period, because of the influence in ΔTECH and ΔOTE, which fluctuated and was smooth, remaining well below 1%. It shows that there is inefficiency during the study period. However, ΔOSME remained above 1% and had a negative correlation with ΔTFP over the study period after a sharp decline from 1.5% to 0.8% in Figure 7b in early 2009 and raising to above 1% after 2010. This means that, in the case of Figure 7a, ΔTFP in the formal sector were mainly driven by both ΔTECH and ΔOTE in early 2009, by ΔOTE during late 2009 and early 2010, and by ΔTECH during early 2012. Additionally, in the case of Figure 7b, by ΔTECH and ΔOTE in 2009, and by the ΔOTE from late 2010 to 2013. In the case of Figure 7c, it was by ΔTECH during late 2009 and during late 2011 and by ΔOTE since early 2009.

In conclusion, ΔTECH and ΔOTE were the most important driving factors that affected ΔTFP negatively over the study period, with the average range lower than 1%. This is in accordance with the results of the annual average (mean) data in Table 6 of the formal sector.

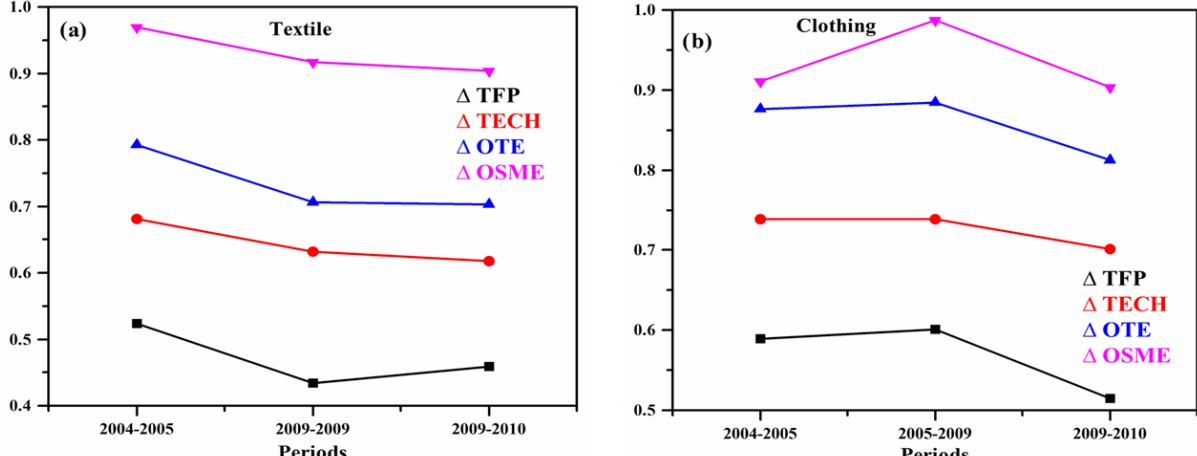

**Figure 8.** The average Hicks–Moorsteen productivity index and its constituents at annual mean over 2004–2005 and 2009–2010. (**a**) Textile; (**b**) Clothing. Notes: ΔOTE: the change in technical efficiency; ΔTFP: the change in total factor productivity; ΔTECH: the change in technological progress; ΔOSME: the change in efficiency–mix and scale (an efficiency value lower than 1 indicates that there is inefficiency in the year of the sector; an efficiency value greater than 1 indicates that there is more efficiency in the year of the sector). Source: Authors' computation.

**Table 6.** Estimations of Hicks–Moorsteen productivity indices of the textile firms (formal sector) for annual means.

|  | Period | ΔTFP | ΔTECH | ΔOTE | OSME |
|---|---|---|---|---|---|
| Very Small Enterprises (VSE) | 2008–2009 | 0.6534 | 0.7629 | 0.8063. | 1.000 |
|  | 2009–2010 | 0.6667 | 0.7182 | 0.7146 | 1.291 |
|  | 2010–2011 | 0.7664 | 0.6518 | 0.7583. | 1.550 |
|  | 2011–2012 | 0.8430 | 0.6224 | 0.8158 | 1.660 |
|  | 2012–2013 | 0.7424 | 0.6891 | 0.7659 | 1.406 |
|  | 2013–2014 | 0.7690 | 0.7224 | 0.7826. | 1.361 |
|  | Mean | 0.7401 | 0.6944 | 0.7738 | 1.371 |
| Small Enterprises (SE) | 2008–2009 | 0.6906 | 0.6270 | 0.7547 | 1.4622 |
|  | 2009–2010 | 0.5789 | 0.7309 | 0.8682 | 0.9139 |
|  | 2010–2011 | 0.6818 | 0.7997 | 0.7793 | 1.0941 |
|  | 2011–2012 | 0.6900 | 0.7088 | 0.8228 | 1.1832 |
|  | 2012–2013 | 0.6379 | 0.6027 | 0.7821 | 1.3534 |
|  | 2013–2014 | 0.7311 | 0.6585 | 0.7067 | 1.5711 |
|  | Mean | 0.6655 | 0.6836 | 0.7839 | 1.2395 |
| Medium Enterprises (ME) | 2008–2009 | 0.7921 | 0.6601 | 0.8723 | 1.1516 |
|  | 2009–2010 | 0.7500 | 0.7107 | 0.8173 | 1.2911 |
|  | 2010–2011 | 0.8314 | 0.6817 | 0.7328 | 1.6642 |
|  | 2011–2012 | 0.8097 | 0.7665 | 0.7172 | 1.4937 |
|  | 2012–2013 | 0.7072 | 0.7316 | 0.6928 | 1.3952 |
|  | 2013–2014 | 0.8522 | 0.7989 | 0.7155 | 1.4908 |
|  | Mean | 0.7888 | 0.7233 | 0.7560 | 1.4048 |
| All textiles firms | Mean | 0.7296 | 0.7002 | 0.7711 | 1.3360 |

Figure 8a,b examines the changes in the share of each type of sub-industry. The informal sector of textile companies Figure 8a revealed a total decline in ΔTFP, ΔTECH, ΔOTE, and ΔOSME throughout the period of study. As regards the clothing companies Figure 8b, the informal sector also reported a decline in ΔTFP, ΔTECH, ΔOTE, and ΔOSME after 2005–2009, which indicated that TFP and its various constituents were mainly potential drivers affecting the informal sector of textile and clothing companies during the period of 2005–2010. Another important observation to notice is that it was the average annual

mean of each constituent was lower than 1%, which meant that there was inefficiency in the years of the sector, which is in accordance with the conclusions of the annual average (mean) data in Table 7 of the informal sector.

**Table 7.** Estimations of Hicks–Moorsteen productivity indices of the textile/clothing firms (informal sector) from annual means.

|  | Period | ΔTFP | ΔTECH | ΔOTE | OSME |
|---|---|---|---|---|---|
| Textile | 2004–2005 | 0.5234 | 0.6813 | 0.7925 | 0.9695 |
|  | 2005–2009 | 0.4338 | 0.6322 | 0.7063 | 0.9717 |
|  | 2009–2010 | 0.4592 | 0.6322 | 0.7029 | 0.9034 |
|  | Mean | 0.4706 | 0.6432 | 0.7722 | 0.9476 |
| Clothing | 2004–2005 | 0.5897 | 0.7388 | 0.8764 | 0.9109 |
|  | 2005–2009 | 0.6010 | 0.7688 | 0.8844 | 0.9873 |
|  | 2009–2019 | 0.5145 | 0.7010 | 0.8125 | 0.9034 |
|  | Mean | 0.5669 | 0.7090 | 0.8571 | 0.9331 |

*6.2. Discussion*

The study investigated the Hicks–Moorsteen TFP index and its constituents. In Table 6, for the whole of the textile companies, there is a decline in the firm productivity of the textile industry with a mean of 72.96%. The drop in firm productivity is thus approximately 27.04%, which is further described by the contemporaneous down in ΔTFP of 25.99% (VSE) and of 33.45% (SE) and of 21.12% (ME). This finding implies an unfavorable frontier shift and leads to increased production costs for a given level of inputs and decrease in TFP. The results also show the decline in firm productivity is more justified by the concurrent drop in ΔTECH of 29.89% and ΔOTE of 22.89%. In contrast, ΔOSME increases up to 3.360 % in the textile sector, but shows insignificant effect in the increase in TFP, indicating that the policy influence on efficiency-mix and scale is weakened, unlike the trends of ΔTECH and ΔOTE, which are the main determination traits in the fall of firm productivity in Cameroon. Based on the above results, (H1) and (H2) are confirmed.

The results are equivalent to those of [74] for Indonesia and [75] in Ukraine. In view of this observation, the effect is twofold: First, TE indicates the ability of the firm productivity to enhance its productivity for a particular stage of inputs. A mechanical result of efficiency of 22.98% thus constitutes the percentage in which the textile sector can increase its capacity without requiring change in the causes of productivity accessiblity. In addition, the decrease in the TE can be described by the challenges associated with the contribution of primary resources. In reality, the contribution of primary resources is sometimes perturbed by frequency insufficiencies of imported primary resources, the significant expenditure of production means, and tariff peaks, which enhance the costs of primary recourses imported. Since there are no guidelines specific to the textile companies as concerns primary resources imports, the controllers of the sector must assemble important financial means to support themselves with enough primary resources [76]. The decline in TE may as well be clarified by the level of the employee potential and the energy consumption cost. Indeed, the domestic labor intensity is low skilled with low income. Apprenticeship and education are secured by specialists of the sector, but the small income offered to the workforce does not enable the growth of reliability and the support of the information gained and training. Therefore, after they obtain the fundamental basis, the trainings adjust on their account with little resources. Additionally, the energy consumption cost is, in the opinion of the controllers of the textile sector, costly and bestows a critical ruin to the effectiveness of the sector. This linked with the regular cutbacks in electricity lead to the lower productivity in the sector; most companies are required to refuge to electrical generator that weighs upon the cost of productivity. Textile industries could also be blamed of traditional organization of productivity and the market data among the firms, which drive to damages in major economies and a raise in costs of production and decline in TFP. Second, the decrease in firm productivity of companies is upheld by the decline in ΔTECH of 29.89%. In fact, TECH

relates to the technology adoption and managerial innovativeness. This finding can be explained by the weak in the research and development (R&D) and public research of the equipment of production of the textile industry in Cameroon, together with the lack of better use of the machinery, servicing, and lower repairs skills. The production equipment is particularly old due to the newly imported equipment, and the rate of devaluation of detentions that reveals the aging of the producing apparatus is settled around 76%; the usually authorized average is fixed at about 48% [76]. Furthermore, there are no existing frameworks of advancement, studies, and evolution of technology to lead, monitor, and train the operatives on their technical decisions. Based on the above conclusions, (H1) is verified.

In Table 7, the survey of TFP of enterprises of the textiles sector shows a meaningful decrease in productivity as a whole. In general, less production defines the operations in the sector. As a matter of fact, ΔTFP is of 47.06%, precisely a fall of 52.94% of productivity. This finding implies that low productivity is described more by the fall in TECH, with ΔTECH fall of 35.68%, and the fall in TE with ΔOTE fall of 22.78%, and the fall in efficiency–mix and scale with OΔSME fall of 3.05%. In the view of the observation, the TFP of the textile companies is mainly driven by TECH, TE, and efficiency–mix and scale over the research period, which is similar to the results of [77,78]. Relying on the results, (H1) and (H2) are confirmed.

In the case of clothing companies, the fall in TFP is very crucial during the period of the study with ΔTFP of 56.69%, greater than for the textile companies, clearly revealing a decrease of 43.31% in productivity. This is justified as concerns the size of clothing companies, which is larger than the size of textile companies. This reveals the slightly improvement of ΔTFP of clothing companies compared to ΔTFP of textile companies. It can be concluded that the firm size of the industry is an important factor positively or negatively affecting TFP due to the experience of larger industries (labor and capital intensives) as the learning by doing effect by [79–81]. This finding implies that the fall in TFP of the clothing companies is explained by the TECH, TE, and efficiency–mix and scale, which is close to the answers of [17,82]. In accordance with the findings, (H1) and (H2) are approved.

For this group of industries, three different types of clothing industries co-exist in the field: A manufacturing mode, a ready-to-wear clothing mode, and a craftsman mode where a large number of industry stakeholders meet. The manufacturing mode, which is usually composed of males (shirts, trousers, and jackets sets) and female (sets including textiles garments) apparel, is increasingly prominent. Occasionally, this kind of productivity is commercialized in specialized shops where it can rival imported goods. Ready-to-wear clothing mode has many challenges regardless of the presence of multiple skills, with the annual rate productivity for a unit being about 25.000 units. Regarding the manufacturing mode and the ready-to-wear clothing mode, productivity capacities are only at around 40% [48].

Overall, the results conclude that firm size, technology adoption, and primary resources (raw material) are the major sources to firm capacity in the production of textiles and clothing industries in Cameroon. This has vital effects on policies to upgrade the applied science and increase the competition and potential of enterprises. Action may be taken to boost R&D business in firms where advanced technology has been extremely slow and motivation can be granted to those with the greatest creative activity.

## 7. Conclusions

This study sought a deeper understanding of the determinativeness of the informal sector and its influence in the firm productivity of textile and clothing industries in Cameroon, SSA, over the period of 2005–10 and 2009–14 using DEA-based and Hicks–Moorsteen index methods to investigate the specific constituents affecting TFP. Additionally, this paper analyzed the effects of the COVID-19 pandemic on the informal sector and the economic situation of Cameroon, and classified determinativeness where developments are needed to boost the informal sector of the leading industries in SSA.

Our findings are twofold. Firstly, the textiles enterprises of the formal sector show a decline in firm capacity. The decline is explained by the fall in TECH and TE. Furthermore, our study revealed that ME has the highest mean of ΔTFP compared to SE and VSE. From the econometric interpretations, we discovered that TECH and TE features play an important role in the calculation of TFP. Our results also explained that the factors determining TFP of companies vary within the sectors. This is in line with the findings from the literature review that firm size, technology adoption, and primary resources (raw material) are the major drivers of firm productivity. Secondly, regarding the informal sector, the overall study of the productivity of textile enterprises showed a significant drop in firm productivity. Concerning the clothing industries, the total drop of ΔTFP is additionally convincing for the entire period. Moreover, we found that the ΔTFP of the clothing industries has the highest mean of TFP compared to the ΔTFP of textile companies due to the size of clothing industries, showing that TFP is not directly related to firm size.

In summary, in connection with (H1) and (H2), these results confirm that TECH, TE, and efficiency–mix and scale negatively affect the TFP of Cameroonian firms.

As it is considered difficult to put an end to informal sector activities, government institutions must therefore provide inducements, as well as advantageous tax treatment, and enhance the accessibility to finance loans to assist them in maintaining employment, since their actions force formal sector industries to more industrialization. National authorities could perhaps envisage a framework of cooperation between formal and informal industries that might allow developing more innovation and thus improve performance and economy growth, along with causing informal enterprises to strengthen. From this point, an important emphasis must be placed on worker training. Indeed, our results show that the labor intensity and capital intensity certify the industrialization capacity of enterprises. In this sense, enterprises must place significant emphasis on importing raw material, technology adoption, and training personnel to expand their skills and talents, and thus take benefit of the opportunities in their environment to promote more innovation.

In the future, as an extension of this research, we shall focus on different strategies for estimating the productivity impacts in sub-industries, such as taking firm productivity as a predictor variable (aleatory variable, state variable, and global variable) and the use of the textiles industry as a uniform item. Nevertheless, this branch of activity incorporates various businesses, from cotton productivity to the garment manufacturing industries.

**Author Contributions:** Conceptualization, methodology, investigation, data curation, visualization, and writing—original draft preparation, N.T.E.; software, writing, and formal analysis, B.I.P.; resources, review and editing, supervision, and project administration, H.C. All authors have read and agreed to the published version of the manuscript.

**Funding:** This research received no external funding.

**Data Availability Statement:** The datasets were generated from The National Institute of Statistics (Chambre de Commerce, d'Industrie des Mines et de l'Artisanat. Textile & Industrie D'habillement) and are available from the National Institute of Statistics, Yaoundé, Cameroun/at contact@stat.cm, with the permission of www.statistics-cameroon.org, accessed on 1 September 2021. Source: Employment and Informal Sector Surveys, EESI, 2005; 2010, General Enterprises Survey, RGE, 2009; 2014.

**Acknowledgments:** Thank you to the African Economic Research Consortium (AERC) for support to carry out this research and the reviewers of the collective book "The Francophone Sub-Saharan African countries facing the Coronavirus (COVID-19): Economic informal sector impacts, response policies, economic crisis and resilience"; published by the Economic Policy Analyst at the Nkafu Policy Institute and Agence Française de Développement (AFD) for their comments and suggestions on an earlier draft of this paper.

**Conflicts of Interest:** The authors declare no conflict of interest.

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
