# Peer review of "The Illustrative Understanding on the Informal Sector and Its Influence in Firm Productivity in Sub-Saharan Africa (SSA): Evidence from Cameroon"

_sustainability, doi:10.3390/su14159789_

Round 1
Reviewer 1 Report
This is an interesting paper that examines if the high degree of the informal sector in Cameroon has any impact on firm level productivity. The authors create a Malmquist index and show that firms of the informal sector in the clothing and textiles sectors witnessed an important decline in their total factor productivity. I have two comments:
1) First, as far as I can see, the authors do not employ any measure of capital stock in their calculations of productivity of textile firms. This is a necessary variable, beacuse capital and labor are two most common inputs in production functions. They need to find a measure of capital to incorporate it in their calculations.
2) The authors need to logically explain why did productivity fell in the infomal firms of Camerron during this period. A plausible explanation is that property rights are ineffective and therefore discourage innovation activity and technology imports.
Author Response
Reviewer 1 : Comments and Responses , Please Check the attache Below.
Thank you.

Reviewer 2 Report
The authors attempt to “evaluate the determinativeness of the informal sector and its impact on firm productivity by applying the Hicks-Moorsteen index and Data Employment Analysis (DEA) methods to estimate factors driving total factor productivity (TFP) and its constituents for Cameroon’s companies over the period from 2005 to 2014”.
I fear the article needs significantly more work before it could be published. I would recommend its publication after major revisions.
The following issues should be addressed:
1. The authors should elaborate more on the notion of informal sector by providing a robust definition of it.
2. It seems that there is a confusion regarding the exact study’s time period. Although the authors inform us that the time period is from 2005-2014, then in the literature review employ more recent data. For what reason?
3. The authors should investigate the impact of SMEs on development. In this vein, see Meramveliotakis, Giorgos, and Manolis Manioudis. 2021. "Sustainable Development, COVID-19 and Small Business in Greece: Small Is Not Beautiful" Administrative Sciences 11, no. 3: 90. https://doi.org/10.3390/admsci11030090 &
de Sousa Jabbour, A. B. L., Ndubisi, N. O., & Seles, B. M. R. P. (2020). Sustainable development in Asian manufacturing SMEs: Progress and directions. International Journal of Production Economics, 225, 107567.
4. There are many claims that need more explanation and bibliography support. For example “In most developing countries from Africa, Asia, Europe, and Latin America, the informal sector is where more than 50% of new jobs opportunities are generated and where the majority of the population, including most of the poor, principally young employees and women”. (40-43)
5. In many instances the English are not the proper. For instance, “The informal sector illustrates the major share of many industries…” (49). And to give another example, “For instance, the disruption of the economic crisis in 2008 and 2020 conducted to a long-term repercussion on the nationwide economy and thus gloomy the advancement of the high technology of firms, which in turn decrease the TFP; the government institution controlling the sector also has possible consequences on TFP” (291-294)
6. Both introduction and literature review should be reduced extensively.
7. Also there is a confusion regarding the final results. At one point the authors claim that “Overall, the results conclude that technology, firm size, labor and capital intensive and intermediate inputs consumption, and competitiveness in the form of imports of primary resources are the major sources drivers to firm capacity in the production of textiles and clothing industries in Cameroon” (805-808) and then argue that “This is in tune with the findings from literature review that raw material, technology adoption, intermediate consumption, firm size and innovation R&D are vital to enhance firm productivity”. Comparing with what the authors include in the abstract I am confused regarding what are the exact determinants of firm’s productivity. What about labor and capital intensive? What exactly the authors mean?
Author Response
Reviewer 2 comments and responses, Please check the attache below.
Thank you.

Reviewer 3 Report
The study explores TFP in textile and clothing formal and informal sectors in Cameroon. The following comments could improve the study.
Regarding Hypothesis 1, are TECH and TE expected to have a positive or negative effects on TFP?
On page 7, it is stated: gloomy the advancement of the high technology of firms, which in turn decrease the TFP. This would suggest that TECH has a positive effect on TFP – more technological progress, higher TFP, and vice versa. However, empirical evidence on page 21 shows the opposite.
In equation 2, what are ρ and δ?
When explaining the results, an explicit link between two hypotheses and empirical findings should be discussed and elaborated. In other words, is hypothesis 1 confirmed by the findings? Is hypothesis 2 confirmed by the findings?
It is not clear why technology progress (TECH) would have a negative effect on TFP (page 21).
It is not clear why technical efficiency (TE) would have a negative effect on TFP (page 21).
Author Response
Please find attached file with responses to reviewer 3 comments.
Reviewer 4 Report
See the attached file.

Author Response
Reviewer 4 , Comments and responses. Please check the attache below.
Thank you.

Round 2
Reviewer 2 Report
The authors should take more seriously my comment, since we spend considerable effort and time for the review.
Point 1: The authors should elaborate more on the notion of informal sector by providing a robust definition of it.
Response 1: Thank you for your suggestion. We have provided more on the definition of informal sector.
I cannot detect any definition provided in the manuscript. Please provide a definitional concept of the term based on the relevant bibliography.
Point 2: It seems that there is a confusion regarding the exact study’s time period. Although the authors inform us that the time period is from 2005-2014, then in the literature review employ more recent data. For what reason?
Response 2: Thank you for your careful comment. In the present, we have clarified the exact study’s time period.
We significantly emphasize on the understanding the economics management evolution of textiles and clothing industries in Cameroon employing the data set from 2005 to 2014. The exact study period is indicated as follows: The first data is the survey of the Employment and Informal Sector (EESI) carried out in 2005 and 2010. The second data is the General Enterprise Survey (RGE) performed in 2009 and 2014 in Cameroon. An eight-year-old dataset is used for reviewing historical documentation on informal sector in Cameroon and their determinativeness and impact on firm productivity. Apart from the use of the data set from 2005 to 2014, exploring a data set (lower eight-year-old dataset or higher) is another important concern.
Authors’ response does not response to my comment. In the literature review (for instace, see p. 9) they analyse more recent data. This should be deteted from the manuscript.
Point 3: The authors should investigate the impact of SMEs on development. In this vein, see Meramveliotakis, Giorgos, and Manolis Manioudis. 2021. "Sustainable Development, COVID-19 and Small Business in Greece: Small Is Not Beautiful" Administrative Sciences 11, no. 3: 90. https://doi.org/10.3390/admsci11030090 &
de Sousa Jabbour, A. B. L., Ndubisi, N. O., & Seles, B. M. R. P. (2020). Sustainable development in Asian manufacturing SMEs: Progress and directions. International Journal of Production Economics, 225, 107567.
Response 3 : Thank you for your suggestion. We have rewritten the manuscript accordingly.
Point 4: There are many claims that need more explanation and bibliography support. For example, “In most developing countries from Africa, Asia, Europe, and Latin America, the informal sector is where more than 50% of new jobs opportunities are generated and where the majority of the population, including most of the poor, principally young employees and women”. (40-43)
Response 4 : Thank you for your careful comment. We have added some references supporting this understanding in the manuscript.
References are:
ILO, “Informal Economy: More than 60 per cent of the world’s employed population are in the informal economy,” 2018, [Online]. Available: https://www.ilo.org/global/about-the-ilo/newsroom/news/WCMS_627189/lang--en/index.htm
IMF, “Five Things to Know about the Informal Economy,” 2021, [Online]. Available: https://www.imf.org/en/News/Articles/2021/07/28/na-072821-five-things-to-know-about-the-informal-economy
Point 5: In many instances the English are not the proper. For instance, “The informal sector illustrates the major share of many industries…” (49). And to give another example, “For instance, the disruption of the economic crisis in 2008 and 2020 conducted to a long-term repercussion on the nationwide economy and thus gloomy the advancement of the high technology of firms, which in turn decrease the TFP; the government institution controlling the sector also has possible consequences on TFP” (291-294)
Response 5 :We are very sorry for wrong writing. We have re-polished the manuscript.
Point 6: Both introduction and literature review should be reduced extensively.
Response 6 : Thank you for your suggestion.
The authors did not reduced the literature review.
Point 7: Also there is a confusion regarding the final results. At one point the authors claim that “Overall, the results conclude that technology, firm size, labor and capital intensive and intermediate inputs consumption, and competitiveness in the form of imports of primary resources are the major sources drivers to firm capacity in the production of textiles and clothing industries in Cameroon” (805-808) and then argue that “This is in tune with the findings from literature review that raw material, technology adoption, intermediate consumption, firm size and innovation R&D are vital to enhance firm productivity”. Comparing with what the authors include in the abstract I am confused regarding what are the exact determinants of firm’s productivity. What about labor and capital intensive? What exactly the authors mean?
Response 7: Thank you for your valuable comment. We have corrected it accordingly.
In. figure 5 it is the “the key determinants” not the “key determinativeness”
Author Response
Reviews 2: Comments and responses.
Reviewer 3 Report
My comments are addressed at a satisfactory level.
Author Response
Reviewer 3: Comments and responses.

Round 3
Reviewer 2 Report
I am now happy with the authors' revisions.